# Reproductive Results in Cancer Survivors after Fertility Sparing Management: The Need for the Standardization of Definitions

**DOI:** 10.3390/cancers15143569

**Published:** 2023-07-11

**Authors:** Szymon Piątek, Iwona Szymusik, Mariusz Bidziński

**Affiliations:** 1Department of Gynecologic Oncology, The Maria Sklodowska-Curie National Research Institute of Oncology, 5 Roentgen Street, 02-781 Warsaw, Poland; bidzinski.m@gmail.com; 2Kazimierz Pulaski University of Technology and Humanities in Radom, 26-600 Radom, Poland; 3Department of Obstetrics, Perinatology and Neonatology, Center of Postgraduate Medical Education, 80 Ceglowska Street, 00-001 Warsaw, Poland; iwona.szymusik@gmail.com

**Keywords:** fertility sparing management, gynecologic cancer, obstetric results, pregnancy rate, birth rate, endometrial cancer, cervical cancer, ovarian cancer, conservative treatment

## Abstract

**Simple Summary:**

The assessment of oncological outcomes has been well defined by overall survival and its surrogates: disease-free/progression-free survival. Although fertility-sparing management (FSM) is used in clinical practice, definitions of reproductive outcomes have not been established. Currently, various definitions are used, and different criteria for the same terms are applied. The aim of this narrative review is to show the diversity in the ways that reproductive outcomes after FSM are reported. It is unknown whether pregnancy or childbirth rates should be the primary endpoints, and the assessment of pregnancy/birth rates is confusing due to the selection of different reference groups. Additional bias is related to “seeking pregnancy” patients, who are distinguished with no clear criteria and are used as a denominator. Moreover, the discussion with patients about the chances of childbearing is complicated. FSM in young women has an unquestionably important role, but uniform definitions of reproductive outcomes should be established.

**Abstract:**

In fertility-sparing management (FSM), two different issues can be distinguished: the risk of recurrence/death and the chance of childbearing. Survival is the principal outcome in oncology, and definitions of overall survival and progression-free survival are therefore well defined and widely accepted. The introduction of FSM to clinical practice was determined by the desire of young cancer patients to still have children. Initially, in small groups of patients, any pregnancy and/or childbirth were considered successes. Nowadays, FSM occupies an important place in cancer treatment, with thousands of young women treated successfully. However, in contrast to survival, no definition has been established for evaluating the reproductive outcomes of FSM. This review article evaluates the current pregnancy and birth rates of cancer patients. Differences between fertility-sparing and conservative treatment are analyzed, and improper and confusing interchangeable applications of these terms are pointed out. Additionally, various reasons for choosing FSM as a treatment method—which are not directly related to fertility preservation (treatment mismatch)—are presented. Uniform definitions of reproduction after FSM should be established to enable the comparison of results and facilitate the counseling of patients regarding the chances of reproduction.

## 1. Introduction

The approach to oncological patients has evolved over the years. Although survival and the reduction of cancer-related mortality will always remain the main goals, the importance of the quality of life of cancer patients has been increasing in recent times. Fertility preservation is one of the factors that can improve the quality of life and diminish the distress of cancer patients [1].

Since the publication of the American Society of Clinical Oncology (ASCO) guidelines in 2006, providing fertility counseling to all cancer patients of reproductive age has been recommended [2]. Oncologists should thus address the risk of fertility impairment prior to gonadotoxic treatment, and they should be prepared to discuss fertility preservation or else refer patients to reproductive specialists. However, a systematic review published in 2022 found that the provision of fertility counseling and fertility specialists’ referral by oncologists treating cancer patients was suboptimal and sometimes unacceptably low, with wide variabilities among the reviewed studies [3]. Nonetheless, the proportion of patients who do not remember any discussion about issues related to fertility prior to treatment is gradually decreasing [4]. Goodman et al. have pointed out that women with breast cancer were 10 times more likely to receive fertility preservation counseling (FPC) compared with other oncological diagnoses [5]. Similar results were obtained by Bastings et al., who reported that women diagnosed with breast cancer or lymphoma were referred to reproductive medicine specialists more frequently compared with other malignancies [6]. Changes are slow, but one can also find rates as high as 81% of cancer patients receiving FPC in the state of Georgia [7]. 

While the involvement of patients in decision-making is growing, it is crucial for patients to be properly informed about both the risks and the benefits of fertility-sparing management (FSM), including reliable data regarding the chance of motherhood. 

Speaking with patients at the time of medical consultation must be as simple as possible in order to be properly understood, but healthcare professionals (both clinicians and nonclinicians) are also obliged to communicate with each other using medical terminology in order to eliminate misunderstandings. Clear definitions in oncology are therefore crucial to understanding the management and the results of the treatment, and they are also vital to comparing the different methods and outcomes between studies. Thus, clinical trial endpoints have been established and are well defined [8]. Overall survival (OS) is considered to be the most reliable cancer endpoint survival, which is defined as the time from randomization until death from any cause. The above endpoint is both precise and easy to measure, and it is documented by the date of death. Bias is not a factor in OS. However, the main limitation of OS is the need for long follow-up, which may be very difficult in cases of early-stage tumors, young patients, or good prognosis malignancies. Surrogate endpoints were therefore established: (1.) disease-free survival (DFS), which is defined as the time from randomization until disease recurrence or death from any cause, and (2.) progression-free survival (PFS), defined as the time from randomization until objective tumor progression or death (whichever occurs first). Although potentially subject to assessment bias, surrogate endpoints are widely used in studies that focus on fertility preservation. 

The effectiveness of reproduction is not less important than survival in fertility-sparing management. Until now, thousands of patients had undergone fertility-sparing treatment due to cervical cancer [9], ovarian malignant tumors [10], and endometrial neoplasms [11], and current recommendations include fertility-sparing methods. Nowadays, FSM is not an experimental treatment, and oncological safety has been proven and confirmed by many independent authors in both metanalyses and systematic reviews [9,10]. While the outcomes regarding survival are similar, the data on the effectiveness of reproduction are heterogeneous. Epidemiological studies have shown that female survivors of cervical and ovarian cancer have the lowest probability of a postcancer pregnancy, with a rate of 6–15% [12]. However, individual authors have presented high reproduction effectiveness in the above malignancies—up to 80% in ovarian cancer [13] or 74% in cervical cancer [14]. These significant differences in outcomes may be caused by the lack of clear definitions and the lack of standardization for assessing reproduction effectiveness.

## 2. Reproductive Definitions in Oncologic Patients: Live Birth Rate, Pregnancy Rate 

Apart from survival, reproductive results are of note in patients undergoing FST. Unlike survival, the methodology and terminology of reproduction data were not specified, and, as a result, various definitions have been used (Table 1). Some authors have avoided using rates, and they have presented the results as crude numbers [15,16]. Sobiczewski et al. defined a successful pregnancy as a pregnancy that ended in childbirth [17]. Fang et al. used the term “successful pregnancy”, but they did not specify whether this term meant clinical pregnancy per se or pregnancy that ended in a live birth [18]. It is obvious that unambiguous definitions facilitate communication between clinicians and enable the comparison of results. Many authors distinguish patients who had undergone FSM as some of the following: “try to conceive”, “attempted pregnancy/to conceive” [14,18,19,20,21,22,23,24,25], “seeking parenthood/pregnancy” [26,27], “desired to get pregnant” [28], “wish to conceive” [29], or “pregnancy desire” [30]. Such described groups of patients in these studies were afterward regarded as denominators in the calculation of pregnancy rates, leading to results as high as 93.3% [31] in endometrial cancer, 89% in ovarian cancer [32], and 74% in cervical cancer [14].

The imprecise definitions of reproductive results were highlighted in two independent systematic reviews conducted by Wei et al. and Schuurman et al. [37,39]. Over 1000 patients with endometrial cancer/atypical endometrial hyperplasia were enrolled in each review. Oncological results were comparable to the studies by Wei et al. and Schuurman et al. [37,39]: a complete response rate of 71% and 80% and a recurrence rate of 20% and 34.7%, respectively. On the other hand, reproduction results differed widely in the studies of Wei et al. and Schuurman et al.: pregnancy rates of 34% and 66.8% and live birth rates of 20% and 70%, respectively. The lower recurrence rate in Wei’s meta-analysis may have been caused by the presence of patients with atypical endometrial hyperplasia (AEH). However, the indications for fertility-sparing treatment in endometrial cancer were strictly defined: endometrial histology and well-differentiated tumors (G1) without myometrium infiltration (IA), and the treatment methods were hormone-based. Therefore, the bias related to different patients’ characteristics or treatment methods was minimal, and oncological outcomes were similar. The different pregnancy rates and live birth rates resulted from different methodologies and probable nomenclature problems. Then, Wei et al. included all patients as the reference group, while Schuurman et al. only included those women who wished to get pregnant.

A wide range of reproductive results was noted by Bercow et al. in patients with borderline ovarian tumors (BOT): pregnancy rates varied from 17.9% to 100%, and live birth rates varied from 23% to 100% [40]. Methods of surgical treatment (cystectomy vs. adnexectomy, unilateral vs. bilateral lesions) and surgical approach (laparoscopy vs. laparotomy) could potentially affect reproductive results, but it has been shown that they did not impact reproductive outcomes [18,20,30]. Not surprisingly, age was related to the ability to conceive [41,42]. No other determinants of fertility after conservative surgery for borderline ovarian tumors, including epidemiological characteristics, preoperative tumor marker levels, tumor size, the type of conservative treatment, and histologic type, were found [41].

Gallo et al., in a review article regarding FSM in endometrial cancer, noted that the pregnancy rate was 47.8%, but that increased to a rate as high as 93.3% when only women “trying to conceive” were considered [43]. Nevertheless, the definition of “trying to conceive” was not precise, so it could lead to selection bias, as each woman and each doctor may understand the term differently. This issue is even more complicated, given the fact that some women are diagnosed with cancer incidentally during infertility diagnostics. Therefore, it is not obvious whether trying to get pregnant means the same for fertile and infertile patients. Ebisawa et al. found that 53.8% of cancer survivors undergoing assisted reproductive technology had an unknown cause of infertility [21].

Identifying a group of patients who wanted to become pregnant has another problem. With such a criterion (“seek for pregnancy”), it is confusing how the results should be interpreted if the patient became pregnant (and/or gave birth), although she did not declare “seeking pregnancy”. The exclusion of patients who did not declare a desire to have a child may impair the assessment of the effectiveness of FSM. It is estimated that 48% of all pregnancies worldwide are unintended [44]. Data regarding unintended pregnancy after FSS are scant. Although having a child seems to be desirable for this group of women, it should not be assumed that every pregnancy after FST is planned and wanted. In a small group of patients after trachelectomy (n = 23), there were two women (8%) who chose to terminate their pregnancies for social reasons [22]. It is uncertain how these women declared their wish for pregnancy. 

As most studies of FSM are retrospective, asking women if they tried to get pregnant after FSM may be inaccurate. Sexuality, fertility, and childbearing may be touchy subjects for cancer survivors, and many factors may influence the responses of the patients. The embarrassment or shame of not being able to conceive can be another issue. Women who did not get pregnant could have denied any attempts to conceive. It was found that different methods of data collection could influence the results, too. Patients may report different things to their doctors than on the questionnaires [45]. In the study by Jin et al., patients were asked to complete a procreation survey at least 6 months after surgery [25]. Only 38 out of 56 patients decided to fulfill the questionnaire, and while as many as 52.6% of the respondents declared their desire to get pregnant, only 17% finally decided to do so. Lastly, it is not known how to manage patients who had gestational surrogacy treatments [36]. 

Stensheim et al. published an epidemiological study on cancer survivors among stage I ovarian and cervical cancer patients [12]. The authors analyzed postcancer pregnancy rates in 27,556 survivors and compared them to those from a matched comparison group from the general population. They revealed the following pregnancy rates: 8–15% for cervical cancer, 6–12% for epithelial ovarian cancer, and 29–32% for germ/sex-cord tumors. Although the study also included older women (age range: 16–45), the rates were significantly lower than in the nonepidemiological studies.

## 3. Fertility-Sparing Management vs. Conservative Management 

Fertility-sparing treatment and conservative management are used interchangeably, mostly in endometrial cancer, but they are not synonyms. The FSM for gynecologic malignancies is composed of operative (fertility-sparing surgery (FSS)) and nonoperative (conservative) treatment methods (Figure 1). FSS preserves the uterus (and at least a portion of the ovary), and the conservative part of FSM mainly relies on hormone-based therapy (systemic or intrauterine device releasing gestagen). However, chemotherapy may also be used as a neoadjuvant systemic treatment in cervical cancer before trachelectomy or as an adjuvant therapy in epithelial and nonepithelial ovarian cancer after FSS (Table 2).

Conservative management (CM) stands for treatment without surgery or other invasive methods. Hormone therapy is most commonly used for early-stage endometrial cancer in patients who desire to preserve fertility, but also in elderly women with contraindications for surgery or comorbidities. Moreover, some patients with contraindications to surgery may not be eligible for hormone therapy due to the lack of estrogen/progesterone receptor expression, the nonendometrial histological type of the tumor, or contraindications to gestagens. In those cases, radiotherapy (external beam radiotherapy +/− brachytherapy) may be used, which is also regarded as a conservative treatment. Although a few cases of successful pregnancy and birth after irradiation of the pelvis in the course of nongynecological cancers have been reported, pelvic radiotherapy does generally lead to infertility [46], and it can therefore be regarded as a conservative, but not a fertility-sparing management. It is known that females treated with pelvic radiation for childhood cancers have an increased rate of uterine dysfunction, too, leading to miscarriages, preterm births, and low birth weight infants. The above complications are related to reduced uterine volume, vessel damage, fibrosis of the myometrium, and endometrial injury. Radiation doses of 14–30 Gy can also lead to irreversible uterine dysfunction in young women [47].

The term “conservative management” was also used in relation to fertility-sparing surgery for ovarian tumors [48,49]. Patients underwent salpingo-oophorectomy on the side of the ovarian tumor, with staging procedures including peritoneal washing, omentectomy, multiple peritoneal biopsies, and pelvic and paraaortic lymphadenectomy. Such extensive surgical procedures should not be identified as conservative treatments.

The age-related barrier is another difference between fertility-sparing treatment and conservative management that is difficult to define. According to the Centre for Disease Control and Prevention (CDC), women of childbearing age are those from 15 to 44 years old [50], while WHO defines women of reproductive age as being between 15 and 49 years of age [51]. Most studies on fertility-sparing treatment refer to women less than 40 years of age, although there are also reports that have an age limit of to 43 years of age [14], 45 years of age [36,52], and even 48 years of age [26]. The above discrepancies may be a product of the fact that it is difficult to deny fertility preservation procedures upon the request of a patient if they are clinically feasible. However, the primary goal must always be to do no harm.

According to the latest recommendations of the European Society of Human Reproduction and Embryology (ESHRE) from 2020 regarding female fertility preservation procedures, all patients require an individual assessment of both the indications and the risks prior to any intervention [53]. It is suggested that for women with overt POI (premature ovarian insufficiency), fertility preservation is not recommended. The above recommendations refer to oocyte/embryo/ovarian tissue freezing [53]. However, such women may want to undergo oocyte donation programs after their malignancy treatment is over, which would be another point on the list of biases if one wants to measure reproduction effectiveness. On the other hand, not all treatment methods impair fertility equally. Therefore, there will always be a balance that needs to be struck between providing fertility preservation to patients at risk, and not providing it when the risk is low [53]. Nevertheless, ESHRE does not point to an age limit with regard to fertility-preserving procedures—all cases are to be managed individually. The same can be applied to fertility-sparing management: the individual approach should be provided by oncologists and surgeons offering FSM, and weighing up the risks and advantages of the above is always a must. 

On the other hand, we must not forget that age is the single most important factor determining female fertility potential. The loss of the ovarian primordial follicle pool is related to a gradual loss of natural fertility, and the accelerated decline is observed in the last 10 years preceding menopause. However, the fertility decline rate is also individual—AMH (anti-Mullerian hormone) secreted by granulosa cells of preantral and antral follicles is representative of so-called ovarian reserve. In addition, oocyte quality is definitely related to age due to both the decline in the quality of oocyte cytoplasm and the increasing genome abnormalities resulting from changes in the formation of the meiotic spindle [54]. 

In 2014, the American College of Obstetricians and Gynecologists provided data on US fertility clinic success rates of IVF procedures according to age. Live-birth rates were as high as 41.5% in women under 35 years of age, declined to 22.1% in those aged 38–40, and further declined to as low as 5% in women aged 43–44 years. At the same time, the rate of miscarriages progressively increased from 11.4% for women aged 33–34 to as high as 36.6% for those older than 42 [55]. Therefore, upon planning FSM, clinicians must always consider both the age and the ovarian reserve of the patient in order to calculate the probable chance of successful assisted reproductive technology in the future.

In conclusion, FSM contains operative (surgical) and nonoperative (conservative) treatment methods, and its aim is to cure. CM is a broad term that includes treatment approaches other than surgery, and that may also be used as palliative management (Figure 2). CM is neither equal to nor equivalent to FSM, and the interchangeable use of these terms should therefore be avoided. 

## 4. Fertility-Sparing Treatment vs. Less Radical Treatment: Treatment Mismatch

The desire to preserve fertility is obviously required for fertility-sparing management. However, in the study of Speiser et al., it should be noted that up to two-thirds of the patients who had undergone successful treatment did not actually seek parenthood [26]. Moreover, some reports include patients who are over 40 years of age. Taking into account oncological treatment and/or the duration of therapy, the chance of pregnancy in those women is negligible, and such management can therefore be controversial [26,36]. With such a low chance of motherhood, the decision for less radical treatment may not be caused by the desire to preserve fertility but may instead result from the fear of complications related to aggressive treatment, especially since the oncological safety of fertility preservation methods has been confirmed [56].

Preferences regarding the extension of surgery were assessed in patients with nonmalignant gynecologic pathologies. It was found that 60% of women would decline a hysterectomy if it did not alter the success of surgery, while 14% of patients would decline a hysterectomy regardless of any benefit [57]. Moreover, approximately 20% of women with pelvic organ prolapse preferred uterine preservation, despite a potentially inferior outcome, due to concerns about having a hysterectomy. The sense that the uterus was important for sexual function was an additional argument for uterine preservation. It was found that women with at least a college education (OR 2.87; 95% CI: 1.08–7.62) and those who believed the uterus was important for their sense of self (OR 28.2; 95% CI: 5.00–158.7) had increased odds of preferring uterine preservation [58]. Reasons for the rejection of a hysterectomy included a sense of feeling older, plus sadness about losing fertility and body image due to the removal of the uterus. Different preferences regarding uterine preservation were found among populations: 31% among Hispanics, 40% among Germans, and 54% among Russians [59,60]. It was found that being older was a positive predictor of the decision to undergo a hysterectomy: compared with younger women below 40 years of age, women aged 45 to 49 years were less likely to use alternative treatments prior to a hysterectomy (OR 0.41; 95% CI: 0.21–0.76) [61]. It is obvious that the age of patients is crucial to making such decisions. First of all, younger patients are usually those who desire further fertility, while older women might have completed reproduction and/or entered the perimenopausal period. Moreover, older cancer patients assume a less active role in making treatment decisions, and they are less likely to collect and analyze all of the relevant information in order to make the optimal decision [62]. On the contrary, young women are actively involved in treatment decisions, and nowadays, they can and do seek information from multiple sources, such as the Internet and social media [63].

The surgeon’s recommendation remains one of the most important factors that influences the choice of treatment. In the study of Sio et al. regarding breast cancer treatment (lumpectomy vs. mastectomy), it was found that 83.1% of patients stated that their surgeon’s recommendation had a major influence on treatment selection options [64]. According to a large study by Letourneau et al. [24] on female reproductive-age survivors with mixed diagnoses, women who received fertility counseling from both the oncology team and the fertility specialist experienced significantly less regret about their decision to preserve fertility than women counseled only by the oncology team [65].

Historically, a mastectomy was the standard of care in the treatment of breast cancer, even in the early stages of the disease. After reports regarding the oncological safety of breast-conserving treatment (BCT), Wilson et al. conducted a study from 1979 to 1987 in which patients with early breast cancer were free to decide about their treatment method (mastectomy vs. BCT) [66]. As many as 35% of younger women (44.7 vs. 49.8 years of age) chose the minimally invasive method, although breast-conserving therapy (BCT) was approved as the preferred treatment for early-stage breast cancer much later, in 1991 [67]. Interestingly, patients who had chosen BCT did not have any main reasons for that choice.

The decision about the treatment method may have resulted from fear: Denberg et al. found that 55% of patients had negative feelings about surgery (or anesthesia) [68], and they believed that the possibility of death on the operating table was real, as well as having (possibly related) doubts about physicians being trustworthy or competent. Some patients also suggested that surgery is messy or haphazard or that surgery can cause a tumor to spread. Patients who rejected radical surgery were also more likely to seriously consider alternative methods [68]. Sociodemographic factors may also play an additional role in treatment choice. Samuel et al. analyzed disparities in the refusal of surgery for three major gynecological cancers (ovarian, endometrial, and cervical), and it was found that age, race, insurance status, and educational attainment may all impact a patient’s decision to either pursue or refuse surgery [69]. Black women and those with the lowest high school graduation rate were more likely to refuse surgery for gynecological cancer, for example [69]. These observations were confirmed in other studies focusing on patients with endometrial cancer who refused surgical treatment [70,71]. Black (RR 2.9; 95% CI 2.1–4.1) and Hispanic (RR 2.0; 95% CI 1.2–3.1) women were more likely to refuse surgery when compared with White patients [71]. Other factors associated with the refusal of surgery included Medicaid insurance, Charlson Comorbidity Index scores of 2 or greater, stage II or III, and whether the patient had received external beam radiation therapy alone. Factors associated with undergoing surgery included an age greater than 41, stage IB, and whether the patient had received brachytherapy [71]. Black patients were more likely to refuse operative management in other cancers as well, including breast, colon, head and neck, and pancreatic types [72,73,74,75,76].

Decisions regarding less radical treatments may not always be related to the desire to have children. Preserving fertility may not be important for every woman who chooses less radical treatment; some patients may declare a strong desire to preserve fertility just to avoid hysterectomy, and apart from preserving fertility, improving the quality of life, limiting complications, and a fear of radical treatment may all significantly influence patients’ decisions about malignancy treatment.

## 5. Conclusions

The importance of FSM in clinical practice is constantly rising in various gynecological tumors, even in selected cases of uterine sarcomas [77]. FSM is an alternative to radical oncological treatment for young cancer patients. Initially, at the time of the implementation of FSM, every pregnancy and childbirth were an argument for the development of this treatment and supported the direction of changes from radical to fertility preserving options. Today, there are thousands of patients who have undergone FSM, and current challenges for the further development of FSM are different than during the introduction of this method. The appropriate assessment of reproductive outcomes is of note, and the precise establishment of the main reproductive endpoint and its surrogates (pregnancy vs. any birth vs. live-term birth) should be discussed by a multidisciplinary team of gynecological oncologists, obstetricians, and specialists in reproductive medicine. This will lead to the further development of FSM and the introduction of new treatment modalities, and it could also enable the comparison of FSM not only in terms of survival but also in terms of reproduction.

## Figures and Tables

**Figure 1 cancers-15-03569-f001:**
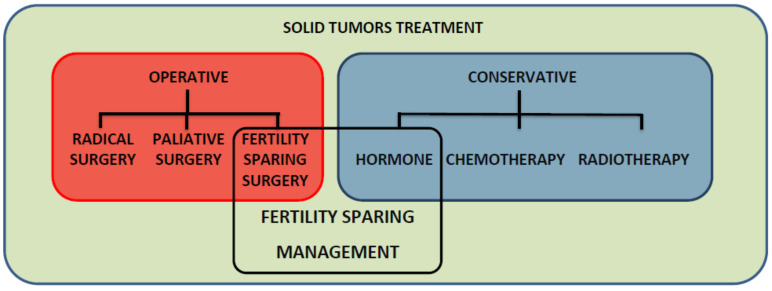
Fertility-sparing management contains both methods of treatment: operative (surgical) and nonoperative (conservative).

**Figure 2 cancers-15-03569-f002:**
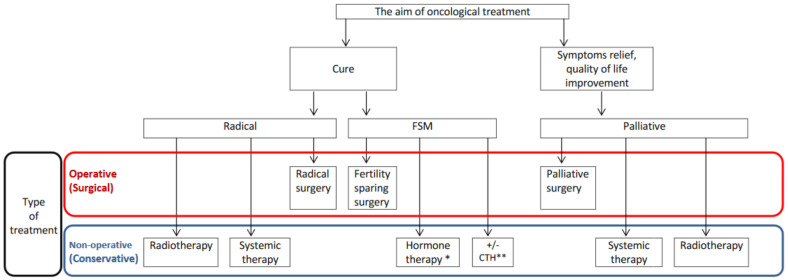
Different oncological management strategies regarding the aim of the treatment and the type of therapy. * systemic gestagen-based therapy or gestagen-relasing intrauterine device. ** adjuvant chemotherapy in FSM may be used in epithelial and nonepithelial ovarian cancer.

**Table 1 cancers-15-03569-t001:** Different terms and criteria for reproductive outcomes assessment after fertility-sparing treatment.

Live Birth Rate	
**Author**	**Definitions**
Capozzi et al. [33]	The number of live-term births divided by the number of pregnancies.
Eskander et al. [34]	The number of live births divided by the number of pregnancies.
Gallos et al. [35]	The number of women who gave birth to healthy infants divided by the total number of women undergoing fertility-sparing therapy.
Peiretti et. al. [19]	The total number of live births divided by patients who attempted to conceive.
Plas et al. [36]	The number of live births divided by the number of pregnancies.
Shepherd et al. [29]	The number of live births divided by the number of pregnancies (excluding patients who terminated the pregnancy).
Wei et al. [37]	The number of women who delivered divided by the total number of patients.
**Pregnancy rate**	
**Author**	**Definitions**
Eskander et al. [34]	The number of pregnancies divided by the total number of patients.
Peiretti et al. [19]	The total number of pregnancies divided by the number of patients who attempted to conceive.
Plas et al. [36]	The number of patients who achieved pregnancy divided by the number of patients who attempted to get pregnant.
Jia et al. [20]	The number of patients who achieved pregnancy divided by the number of patients who tried to conceive.
**Author**	**Other terms and definitions**
Fang et al. [18]	**Pregnancy outcome**—The number of patients achieving pregnancy divided by the number of patients attempting to get pregnant.
Johansen et al. [38]	**Conception rate**—The number of patients with childbirth after surgery divided by the patients treated with fertility-sparing surgery.
Shepherd et al. [29]	**Cumulative pregnancy rate**—Based on the women who desired to conceive as a denominator using the Kaplan–Meier method. **Cumulative actuarial probability of conception adjusted for contraceptive usage**—Proposed for women who are trying to conceive following successful treatment (the Kaplan–Meier method).

**Table 2 cancers-15-03569-t002:** Treatment methods of FSM in gynecological cancers.

	Treatment Method	
Cancer Type	Operative (Surgical)	Non-Operative (Conservative)
Cervical cancer	+	− *
Endometrial cancer	−	+ (Hormone based therapy)
Borderline ovarian tumor	+	−
Ovarian/Fallopian tube cancer	+	−/+ **
Nonepithelial ovarian cancer	+	+ (Chemotherapy)
Gestational trophoblastic neoplasia	−	+ (Chemotherapy)

*—in tumors greater than 2 cm, neoadjuvant chemotherapy may be discussed with the patient, but it should not be considered as a standard treatment. **—adjuvant chemotherapy in high-risk stage I (stage IA grade 2–3, stage IC any grade, clear cell histology).

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
