# Peer review of "Reproductive Results in Cancer Survivors after Fertility Sparing Management: The Need for the Standardization of Definitions"

_cancers, 2023, doi:10.3390/cancers15143569_

Round 1
Reviewer 1 Report
A review paper evaluates current terms of pregnancy rates and birth rates. FSM as a treatment method offers the possibility of childbearing.
1. The authors describe the importance of FSM and its advantages compared to conservative management. A good explanatory essay, but not a review paper.
2. Confusing language expressions. For example, Page 1, the last two sentences in the summary, ‘Therefore, comparison of study results is complicated, whereas discussion with patients about the chances for childbearing is very difficult. FSM in young women plays an unquestionable role, however uniform definitions of reproductive outcomes should be established.’ I didn't get any useful information from the summary.
A brief explanation of other details, such as:
3. The abstract states that the review evaluates the current terms of pregnancy rates and birth rates, which is too vague and imprecise. Based on the subsequent text, it would be more appropriate to add modifiers, such as the pregnancy rates and birth rates of cancer patients.
4. Is the healthcare professional mentioned in the third sentence of the second paragraph of the introduction the same as the one mentioned in the second sentence of the fourth paragraph below? If so, the clarification in parentheses (both clinicians and non-clinicians) should be added when it first appears.
5. The first sentence of the second section of the article is not fluent, and a comma should be added after apart from.
6. In the fifth paragraph of the second section (2. Reproductive definitions in oncologic patients: live birth rate, pregnancy rate), the first sentence mentions another problem, but it is not explained below. What is this problem?
7. In the fourth paragraph of the second section (2. Reproductive definitions in oncologic patients: live birth rate, pregnancy rate), the abbreviation ‘FSS’ is used in the first sentence, but it is not defined. What does it stand for?
8. In the sixth paragraph of the third section (3. Fertility-sparing management vs conservative management), the abbreviation ‘ART’ is used in the last sentence, but it is not defined. What does it stand for?
9. In the seventh paragraph of the third section (3. Fertility-sparing management vs conservative management), it is stated that conservative management (CM) includes fertility sparing management (FSM). However, in the first paragraph of this section, it is stated that ‘Fertility-sparing treatment and conservative management are used interchangeably’. This creates confusion about the relationship between these terms. Please clarify it.
10. The fourth section (3. Fertility-sparing treatment vs less radical treatment: treatment mismatch) is numbered incorrectly. It should be (4. Fertility-sparing treatment vs less radical treatment: treatment mismatch). The subsequent sections should be renumbered accordingly.
11. In the fifth paragraph of the fourth section (4. Fertility-sparing treatment vs less radical treatment: treatment mismatch), can you briefly explain how social geographic factors affect the choice of treatment?
12. Besides the above problems, there are many paragraphs where the content is not coherent, the word order is incorrect, and the reading is not smooth, making it hard to understand.
This manuscript is not qualified to be published on Cancers in this current form.
The review is well-written, but it could be more coherent. It would be helpful to use some expressions of causality and logical relationship to connect the ideas and arguments.
Author Response
Thank you very much for your time and valuable comments. We tried to address them all below step by step. We hope you will find our explanations and corrections satisfactory.
Ad. 1.
Cancers journal allows for two types of reviews: narrative and systematic. Our article is submitted as a narrative review. According to the narrative review criteria of Cancers journal, we tried to objectively report the current state of knowledge on fertility sparing management (FSM) based on previously published research, identify gaps and controversies.
Our article also tries to identify directions for future research (especially the establishment of uniform definitions in FSM). We do not only restate information from other reports, but try to advance the understanding of a reproductive management among cancer survivors.
We also tried to critically comment results of existing literature, especially with regard to the FSM success rated.
Ad 2.
We tried to improve the simple summary text. We believe that simple summary contains several useful information for the readers:
1) Lack of uniform definitions of reproductive results in studies which analyze FSM
2) Different criteria for birth/pregnancy rates are used in studies focusing on FSM
3) Survival and reproduction are assessed in FSM studies. In relation to survival: the primary aim is the overall survival. With regard to reproduction it is unknown what should be assessed as the main goal.
4) Due to different criteria for birth/pregnancy rates used in published studies, the comparison of reproduction results is complicated.
5) There is a need to establish definitions for patients after FSM
We changed some sentences in the simple summary – all the changes are marked in the text. We hope you will find the changes adequate and satisfactory.
Ad 3.
Thank you very much for pointing out what is imprecise. We added “pregnancy rates and birth rates of cancer patients” in the text of the abstract. We hope it makes it more clear now. The change is marked in the text.
Ad 4.
The meaning of “healthcare professionals” is different in these sentences. In the first sentence, we mean doctors (clinicians) who directly manage patients with cancers. In the second sentence “healthcare professionals” include all employees related to health care (doctors, psychologists, midwives, nurses, medical assistants, laboratory technicians, managers and investigators).
To avoid misunderstanding we changed the first sentence as follows:
However, a systematic review published in 2022 found that the provision of fertility counselling and fertility specialists’ referral by treating oncologists was suboptimal, sometimes unacceptably low, with wide variabilities among reviewed studies.
The second sentence remains unchanged, as we defined it in parenthesis (clinicians and non-clinicians).
Ad. 5.
Thank you for noticing that. We added a comma after the word “survival”. This is how the sentence looks now:
Apart from survival, reproductive results are of note in patients undergoing FSM.
Ad 6.
We would like to point out an additional problem related to the isolation of a group of women, who are seeking pregnancy. With such a criterion (“seek for pregnancy”), it is confusing how the results should be interpreted, if the patient became pregnant (and/or gave birth), although she did not declare for “seeking pregnancy”. The exclusion of patients who did not declare a desire to have a child and give birth, may impair the assessment of the effectiveness of FSM.
In the whole population of women worldwide, unwanted pregnancies account for as much as 48%. Data on women with unwanted pregnancy after FSM are sparse. It can be assumed that women after FSM decide to have children more consciously and in a more predictable way, but we lack reliable scientific proof.
Ad 7.
In order to standardize the abbreviations we followed other Reviewer’s comments and changed fertility sparing surgery to fertility sparing management (FSM). We changed FSS/FST into FSM to make the text easier to understand for the reader. We explained that FSM consist of 2 treatment types: operative (FSS-fertility sparing surgery) and non-operative (conservative).
We also added a figure explaining all kinds of treatment of solid tumors (according to the Editorial Office request) – the figure is also clarifying the use of different terms. We hope you will also find it satisfactory.
Ad 8.
The abbreviation ART stands for assisted reproductive technology – it was added in the text and the sentence was changed as follows:
Prior to modification: Therefore, upon planning FSM, clinicians must always consider both age and ovarian reserve of the patient, in order to calculate probable chances of successful ART in the future.
Sentence after modification:
Therefore, upon planning FSM, clinicians must always consider both age and ovarian reserve of the patient, in order to calculate probable chances of successful assisted reproductive technology in the future.
Ad 9.
Thank you for pointing that part out. Indeed, fertility-sparing treatment and conservative management are used interchangeably in the literature, although they SHOULD NOT BE. This is the problem we are trying to address in our manuscript. In order to show why those terms are mistakenly used interchangeably we added a figure (as mentioned before). It clearly shows the types of treatment of solid tumors. We also added a few sentences to the aforementioned paragraph:
FSM of gynecologic malignancies is composed of operative (fertility sparing surgery - FSS) and non-operative (conservative) treatment methods (Figure 1). FSS preserves the uterus (and at least a portion of the ovary). Conservative part of FSM mainly relies on hormone-based therapy (systemic or intrauterine device releasing gestagen). However, chemotherapy may also be used as neoadjuvant systemic treatment in cervical cancer before trachelectomy or as adjuvant therapy in non-epithelial ovarian cancer after FSS.
In order to precisely show different kinds of fertility sparing treatment of gynecological cancers Table 2 was introduced in the text:
Table 2. Treatment methods of FSM in gynecological cancers.
|
|
Treatment method |
|
|
Cancer type |
Operative (surgical) |
Non-operative (conservative) |
|
Cervical cancer |
+ |
- * |
|
Endometrial cancer |
- |
+ (hormone based therapy) |
|
Borderline ovarian tumor |
+ |
- |
|
Ovarian/Fallopian tube cancer |
+ |
-/+ ** |
|
Non-epithelial ovarian cancer |
+ |
+ (chemotherapy) |
|
Gestational trophoblastic neoplasia |
- |
+ (chemotherapy) |
* - in tumors greater than 2 cm neoadjuvant chemotherapy may be discussed with the patient, however, it should not be considered as a standard treatment.
** - adjuvant chemotherapy in high-risk stage I (stage IA grade 2–3, stage IC any grade, clear cell histology)
In conclusion, FSM contains operative (surgical) and non-operative (conservative) treatment methods, and its aim is to cure. CM is a broad term that includes treatment approaches other than surgery, and that may also be used as palliative management (Figure 2). CM is neither equal to nor the equivalent of FSM, and the interchangeable use of these terms should therefore be avoided.
Table 2 summarizes treatment methods in FSM of gynecologic cancers.
Ad 10.
Thank you for pointing that out. The numbering of sections is corrected throughout the text.
Ad. 11.
In order to address that comment we tried to elaborate more on that subject in the text. We also added some new literature to support that information.
Samuel et al. analyzed disparities in refusal of surgery for 3 major gynecologic cancers (ovarian, endometrial, cervical). It was found that age, race, insurance status and educational attainment may impact a patient's decision to pursue or refuse surgery. Black women and those with poorer educational status were more likely to refuse surgery of gynecologic cancer. These observations were confirmed in other studies focusing on patients with endometrial cancer, who refused surgical treatment. Black (RR 2.9; 95%CI 2.1 to 4.1) and Hispanic (RR 2.0; 95% CI 1.2 to 3.1) women were more likely to refuse surgery when compared with White patients. Other factors associated with refusal of surgery included: Medicaid insurance, Charlson Comorbidity Index scores of 2 or greater, stage II or III, receiving external beam radiation therapy alone. Factors associated with undergoing surgery included: age greater than 41, stage IB, receiving brachytherapy.
Black patients were also more likely to refuse operative management in other cancers, including breast, colon, head and neck, pancreatic.
Ad 12.
In order to make the text coherent and understandable we did our best to correct / clarify / simplify as many sentences as possible. We also had the manuscript revised by a native speaker. We hope you will find it improved and satisfactory.
Once again thank you very much for your time and valuable comments.
Reviewer 2 Report
Authors summarized the diversity of fertility sparing management and concluded the necessity to establish the criteria for fertility sparing management. This review is well organized and written. The numbering of "Fertility-sparing treatment vs less radical treatment: treatment mismatch" and "Conclusions" was not correct. It would be better if authors could add more tables or figures in section 3 and 4.
Author Response
Thank you very much for your time and valuable comments. We tried to address them all below step by step. We hope you will find our explanations and corrections satisfactory.
Thank you for pointing out wrong numbering of sections. The numbering of sections is corrected throughout the text.
We added sentences to the main text, one table and two figures. After revision, our manuscript contains > 4000 words, 2 figures and 2 tables.
In addition, our manuscript was revised by a native speaker (MDPI Author Service).
Once again thank you very much for your time and valuable comments. We hope you will find it suitable for publication in Cancers.

Reviewer 3 Report
Manuscript sent to me for evaluation entitled " Reproductive results in cancer survivors after fertility sparing -management: the need for the standardization of definitions” I have read with great interest. The text addresses the very topical issue of female cancer survivors following fertility-sparing treatment.
This manuscript provides a very interesting overview of the current state of knowledge on fertility after cancer treatment. This manuscript, in my opinion, presents a great educational value for the potential reader. As a reviewer, however, I do have one comment - the Conclusions section is too long. I suggest that this section of the manuscript should be rewritten to summarise in a more synthetic way the very interesting analysis of the FSM problem.
In my opinion, the text, after a cosmetic improvement of the Conclusions section, fully deserves to be published.
Author Response
Thank you very much for your time and valuable comments. Thank you very much for your time and valuable comments. We especially thank you for pointing out the educational value of our manuscript. However, together with the reviews, we received information from the Editorial Office to add more words into the conclusion part. Therefore, we decided not to shorten that part of the manuscript. I hope such an explanation is satisfactory for you. Once again thank you very much for your time.
In addition, our manuscript was revised by a native speaker (MDPI Author Service).
Once again thank you very much for your time and valuable comments.

Reviewer 4 Report
The authors reviewed previous reports on gynecological patients who underwent fertility preservation management and discussed issues in those reports. Among several important points, the authors point out the diversity of the patient population and the definition of outcomes. In general practice, it is often experienced that patients receiving fertility preservation treatment do not necessarily wish to become pregnant immediately.
The fact that the target population is all gynecological cancer patients desiring fertility preservation treatment makes the issue somewhat ambiguous. In fact, there are differences in the patient impact and oncologic/pregnancy outcomes of fertility-sparing treatment in cervical, uterine, and ovarian cancers.
The use of three different terms in the text, Fertility-sparing management (FSM), fertility-sparing surgery (FSS), and fertility-sparing treatment (FST), may also make it somewhat difficult to understand. Reviewers believe that the authors need to address the definitions of these terms.
No major problems, but some errors need to be corrected, such as two periods in L319.
Author Response
Thank you very much for your time and valuable comments. Indeed, we wanted to point out the diversity of the patient population and – what is most important – the diversity of terms used in the literature with regard to pregnancy outcomes in gynecological cancer survivors. Our goal, among others, was to show how difficult it is to properly present the rates of successful fertility sparing management.
We agree that using three different terms for fertility sparing (FSM, FSS, FST) is not easy for the readers. To make the text more understandable we decided to use one term: FSM – fertility sparing management. The changes were introduced throughout the main text and marked properly. We also hope that the figure we added to the manuscript (following the Editorial Office request) will clarify everything. It presents the types of treatment of solid tumors and FSM is highlighted there also.
In order to precisely show different kinds of fertility sparing treatment of gynecological cancers Table 2 was also introduced in the text.
We hope you find all our changes satisfactory. Thank you very much for your time once again.
